# Dynamic PET Image Generation from Analytically Derived Time Activity Curves fitted with Physics-Informed Neural Networks

**Shu-Jiun Lin**                                                    R13528005@NTU.EDU.TW

**Yu-Nong Lin**                                                    LINYUNONG@NTU.EDU.TW

**Kevin T. Chen**                                                    CHENKT@NTU.EDU.TW

*No. 1, Sec. 4, Roosevelt Rd., Taipei 106319, Taiwan (R.O.C.)*

## Abstract

Dynamic PET (dPET) enables the assessment of spatiotemporal aspects of tracer distribution to evaluate complex metabolic processes. The two-tissue compartment model (2TCM) (Rizzo et al., 2019) is commonly used for dPET analysis, but it requires an arterial input function (AIF), which must be measured by arterial blood sampling (Muzi et al., 2012). By solving the ODEs derived from the 2TCM (Rizzo et al., 2019) and Feng's model (Feng et al., 1993), we found the analytic solutions of the time-activity curves (TACs), which were integrated with a U-Net to form a physics-informed neural network (PINN) structure. After training the PINN on $[^{11}C]$PIB and $[^{18}F]$FDG PET/MR images, the model was evaluated on the test set using $R^2$ and SSIM. The resulting model enables denoising, kinetic parameter estimation, non-invasive AIF retrieval, and flexible temporal interpolation/extrapolation.

**Keywords:** AIF, dPET, Simultaneous Estimation, 2TCM, Unet.

## 1. Introduction

Dynamic positron emission tomography combined with the two-tissue compartment model (2TCM) (Rizzo et al., 2019) to quantify radiotracer kinetics and provide insights into complex tissue-specific metabolic processes. Traditionally, 2TCM requires an arterial input function (AIF), which necessitates invasive arterial cannulation (Muzi et al., 2012). To reduce this need, this study followed recent physics-informed trends (Guo et al., 2023; Ran et al., 2026) and proposes a physics-informed neural network (PINN) to generate the AIF and parameterize the dPET data. We derived the theoretical analytic solution for the TACs by combining the 2TCM (Rizzo et al., 2019) with Feng's model for the AIF (Feng et al., 1993). This analytical solution is embedded in our U-Net model, enabling it to estimate the 2TCM (Rizzo et al., 2019) parameters for each voxel in the dPET series, as well as the parameters of Feng's model (Feng et al., 1993).

## 2. Materials and Methods

The dataset used for this study comprises 88 $[^{11}C]$PIB PET/MR images from 88 patients (training:72, validation:8, test:8) and 67 $[^{18}F]$FDG PET/MR images from 59 patients (training:47, validation:6, test:6) with various diagnoses, including male and female individuals of different ages. Each dataset contains 1 MRI frame and multiple dPET frames (PIB:

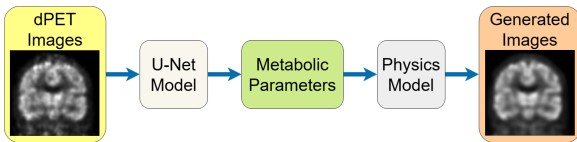

Figure 1: Basic PINN structure

23; FDG: 33). The basic design of our PINN structure is shown in Figure 1, in which the physics model is the combination of 2TCM (Rizzo et al., 2019) and Feng's model for AIF (Feng et al., 1993). We get the theoretical variation curve:

$$c_{measured}(t) = z_1\{(i_1\overrightarrow{e_{\omega 2}} + j_1\overrightarrow{e_{\omega 3}}) + \gamma_1\overrightarrow{\kappa_{121}} - (\alpha_1\overrightarrow{\kappa_{11}} + \alpha_2\overrightarrow{\kappa_{21}} + \alpha_3\overrightarrow{\kappa_{32}} + \alpha_4\overrightarrow{\kappa_{43}})\} + V_b c_b(t) \quad (1)$$

where $z_1$, $i_1$, $j_1$, $\alpha_1$, $\alpha_2$, $\alpha_3$, $\alpha_4$, $\overrightarrow{e_{\omega 2}}$, $\overrightarrow{e_{\omega 3}}$, $\overrightarrow{\kappa_{121}}$, $\overrightarrow{\kappa_{11}}$, $\overrightarrow{\kappa_{21}}$, $\overrightarrow{\kappa_{32}}$, and $\overrightarrow{\kappa_{43}}$ are determined by the parameters in 2TCM and Feng's model and all the parameters shown as vectors are time series. The relationships between parameters in Equation 1 with the 2TCM and Feng's model parameters are not shown in this work due to space constraints. Note that the time points of all time series can be arbitrary. By adjusting the desired time points, we can perform model-based interpolation and extrapolation for resampling and future-frame prediction.

A modified U-Net model was designed to generate the desired parameters from the dPET images. By using the generated parameters as the final activation function in Equation 1, our model can produce a regression line that fully complies with 2TCM and Feng's model. To train our models with 2 types of tracers under different time tables, we also designed a shared-core bidirectional double-model training structure (BDDMTS, Figure 2) to ensure cross-tracer generalizability and temporal consistency between PIB and FDG datasets. In addition to the bidirectional model design, we also implemented a shared core design across the two models. The 2 U-Nets share a common core with identical learnable weights throughout training. By imposing this restriction, we forced the shared core to generalize across both tracer types and time tables.

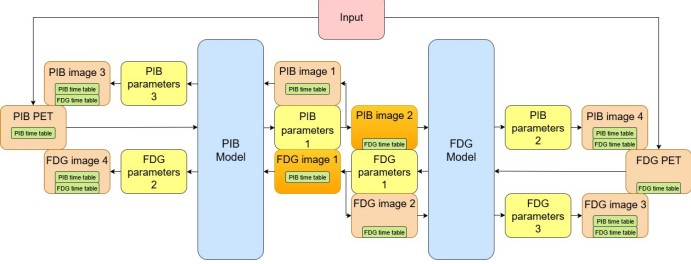

Figure 2: Bidirectional double-model training structure.

During training, the total loss function is the weighted sum of the $MSE$ loss between the original dPET images and generated dPET images, the $L1$ loss between the generated parameters, and the $MSE$ $V_b$ range loss that forces $V_b$ to range from 0 to 1. The model was

trained using PyTorch with the Adam optimizer, starting with a learning rate of $10^{-4}$ and decaying exponentially to $10^{-6}$ in 4000 epochs. For evaluation, the $R^2$ value between the generated signals and the original images is computed. We further used SSIM (Structural Similarity Index) as an additional assessment to reflect human visual perception.

## 3. Results

The best model during training was found at epoch 3439 based on the validation loss. The comparison of the original and generated data is shown in Figure 3. Test data showed $R^2$ value of $0.9737\pm0.0097$ (mean $\pm$ std) for $[^{11}C]$PIB tracer and $0.8812\pm0.0145$ for $[^{18}F]$FDG. In terms of SSIM, our model showed $0.9375\pm0.0144$ for $[^{11}C]$PIB tracer and $0.7908\pm0.0150$ for $[^{18}F]$FDG tracer. These results indicate strong model fit and therefore suggest good estimation of metabolic parameters. Combining the estimated parameters with Equation 1, we accomplished four key objectives: (a) denoising, (b) retrieving physiological parameters, (c) non-invasive AIF estimation, (d) generating a 4D image with flexible temporal interpolation/extrapolation.

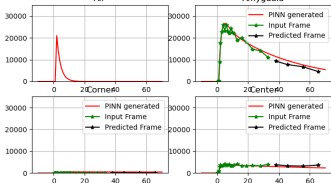

Figure 3: Original and generated $[^{11}C]$PIB PET image. Left: AIF and TAC of specific voxels. Right: Original (left) and generated image at $t = 33.0$ (up, the last frame input into the mode) and 65.5 (down, last frame scanned) minutes.

## 4. Conclusion

This work presents an effective U-Net model for estimating $k_i$ and AIF. Our model maintains strong reconstruction ability after parameterization and removes Poisson noise. Future work includes expanding the model's generalizability to other radiotracers.

## Acknowledgments

This work was financially supported by the Yushan Fellow Program (NTU-114V1015-5) and the Higher Education Sprout Program ("Center for Advanced Computing and Imaging in Biomedicine, NTU-114L900703" from the Featured Areas Research Center Program and the National Taiwan University Career Development Project, NTU-115L7848, NTU-115L7770) of the Ministry of Education, National Science and Technology Council (114-2628-E-002-019-MY3), and National Health Research Institutes, R.O.C. Taiwan (NHRI-EX115-11205EC).

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
