# OpenReview forum: "Dynamic PET Image Generation using a Physics-Informed Neural Network with Analytically Derived Time Activity Curves"
_MIDL.io/2026/Short_Papers — MIDL 2026 - Short Papers Poster_

### Official Review · Reviewer_H9GM · 2026-04-29
**Interesting ideas but need more clarity of details**

**Rating:** 2
**Confidence:** 4

**Review:**

Quality: Nice motivation of methodology with PINN for dynamic PET parametric analysis. But needs more method/experimental details.

Clarity: Many aspects of the paper require more clarity (see weaknesses).

Originality: Prior work in incorporating physical models to dynamic PET analysis should be acknowledged/discussed. Just a couple example:
Ran et al., "Reinforced physiology-informed learning for image completion from partial-frame dynamic PET imaging", MedIA 2025.
Guo et al., "Mcp-net: Introducing patlak loss optimization to whole-body dynamic pet inter-frame motion correction", TMI 2023.

Significance: A framework that allows for accurate analysis of parametric PET without the need for arterial blood sampling would be impactful.

**Summary:**

This paper proposes applying physics informed neural network model for dynamic PET parametric analysis and image generation. The method combines the Feng model for arterial input function with the two-tissue compartment model, and parameters generated by the network are used as part of the loss function in training the network. The method is evaluated on 2 tracers for image generation quality.

**Strengths:**

1. The work aims to incorporate physical models to guide learning of a generation model for dynamic PET data. The use of known physical models that describe PET tracer kinetics could help to constrain the dynamic PET image generation.

2. Providing a framework that does not require arterial blood sampling for parametric PET modeling would be impactful.

3. The work evaluates the methods on two different tracers.

**Weaknesses:**

1. More methods details needed:
While I understand this is a short paper, more details are needed for the reader to understand the approach.
- For example, eq. 1 has numerous variables, which authors state they do not explain/define due to limited space. I can understand not deriving the expression, but at least the variables need to be defined.
- Fig. 2 is hard to understand. I try to follow the flow of the arrows, but cannot understand why certain blocks flow into others. For example, I don't understand why a PiB image goes into the FDG model - what do these have to do with one another (this may be due to my lack of knowledge about the tracers?) Even with the accompanying main text, I cannot quite make sense of what the bidirectional double model training is really doing.
- What is V_b range loss?

2. More experimental methods details needed:
While I again understand this is a short paper, there are important details missing and analysis missing.
- how is data divided for training/testing (and validation)?
- Is the R^2 computed using the whole sequence of voxels and across the dynamic image sequence, or average of each voxel time series? I imagine this would be very high if using the whole sequence of generated data.
- It would be informative if the paper had some evaluation of the estimated AIF parameters - I realize this would require a ground truth input function to be available, but this is better than evaluation based on image quality.
- Another informative experiment would be to compare to parameters estimated by standard fitting of the physical model, using whatever the standard is currently for image-derived input function (if blood samples are not comparable). This would help show potentially the advantage or need for setting this up as an image generation problem.

3. Statements are not supported by results:
Authors suggest that the SSIM of ~0.57 for FDG suggests a "strong model fit" and thus "good estimation of metabolic parameters". I would argue that SSIM of .57 is not such high fidelity (as compared to PiB SSIM of ~.82).  Also, I am not sure about the concluding claim of accomplishing "denoising", as evaluations are all compared to the original noisy signal - that is, I don't see measures for denoising performance.

**Justification Of Rating:**

I like the idea of using physical models to support neural network learning and dynamic PET analysis. However, the paper currently needs more clarity in description of methods and experiments and is hard to follow as is. Furthermore, other evaluations against a reasonable baseline would have been informative for assessing the goodness/advantage/usefulness of the proposed approach.

---

### Decision · Program_Chairs · 2026-05-08

Accept (Poster)